# Synergistic Effects of Anthocyanin-Enriched *Morus alba* L. Extract and Vitamin C: Promising Nutraceutical Ingredients in Functional Food Development for Neuroprotection

**DOI:** 10.3390/foods14213630

**Published:** 2025-10-24

**Authors:** Nootchanat Mairuae, Jinatta Jittiwat, Kwanjit Apaijit, Parinya Noisa, Gang Bai, Yuanyuan Hou, Nut Palachai

**Affiliations:** 1Faculty of Medicine, Mahasarakham University, Mahasarakham 44000, Thailand; nootchanat.m@msu.ac.th (N.M.); jinatta.j@msu.ac.th (J.J.); kwanjit.s@msu.ac.th (K.A.); 2School of Biotechnology, Institute of Agricultu ral Technology, Suranaree University of Technology, Nakhon Ratchasima 30000, Thailand; p.noisa@sut.ac.th; 3State Key Laboratory of Medicinal Chemical Biology, College of Pharmacy and Tianjin Key Laboratory of Molecular Drug Research, Nankai University, Tianjin 300353, China; gangbai@nankai.edu.cn (G.B.); houyy@nankai.edu.cn (Y.H.)

**Keywords:** *Morus alba*, vitamin C, oxidative stress, neuroprotection, hydrogen peroxide, SH-SY5Y cells, apoptosis, MAPKs, synergistic effect

## Abstract

Oxidative stress-induced mitochondrial dysfunction and apoptosis are critical factors in the pathogenesis of neurodegenerative diseases. This study investigated the synergistic neuroprotective effects of anthocyanin-enriched *Morus alba* L. extract combined with vitamin C (MAC) against hydrogen peroxide (H_2_O_2_)-induced oxidative stress in SH-SY5Y neuronal cells. Exposure to H_2_O_2_ triggered excessive reactive oxygen species (ROS) production and apoptosis, whereas treatment with MAC markedly alleviated these effects. Biochemical analyses revealed that MAC significantly reduced malondialdehyde (MDA) and enhanced the activities of antioxidant enzymes, including catalase (CAT), superoxide dismutase (SOD), and glutathione peroxidase (GSH-Px), thereby contributing to improved redox balance. Furthermore, MAC modulated apoptosis-related signaling by upregulating extracellular signal-regulated kinase (ERK), cAMP response element-binding protein (CREB), and the anti-apoptotic protein B-cell lymphoma 2 (Bcl-2), while downregulating the pro-apoptotic protein Bcl-2-associated X (BAX) and cleaved caspase-3. Collectively, these findings demonstrate that MAC acts synergistically as a promising nutraceutical ingredient, supporting the development of functional foods for the prevention or mitigation of oxidative stress-related neurodegenerative disorders.

## 1. Introduction

Neurodegenerative diseases, including Alzheimer’s disease (AD) and Parkinson’s disease (PD), represent a growing global health concern due to their increasing prevalence in aging populations. As of 2023, more than 55 million people worldwide are living with dementia, with Alzheimer’s disease accounting for approximately 60–70% of cases, a number projected to reach 78 million by 2030 and 139 million by 2050 [1]. Similarly, Parkinson’s disease affects over 10 million people globally, with incidence rising rapidly in older adults [2]. These disorders are characterized by progressive neuronal loss, cognitive and/or motor decline, and reduced quality of life, placing a substantial socioeconomic burden on families, caregivers, and healthcare systems [3]. The rising prevalence, limited treatment options, and irreversible nature of neuronal damage underscore the urgent need for research into neuroprotective strategies that can mitigate disease progression or delay onset.

One of the key pathological mechanisms involved in neurodegeneration is oxidative stress, which leads to cellular injury through excessive production of ROS [4,5,6]. Oxidative stress disrupts redox homeostasis, damages cellular macromolecules, impairs mitochondrial function, and ultimately triggers apoptosis [7,8]. In particular, hydrogen peroxide (H_2_O_2_) is a major contributor to oxidative damage and is widely used as a model oxidant in neuronal cell studies [9,10,11]. The mitogen-activated protein kinase (MAPKs) signaling cascade, including extracellular signal-regulated kinase (ERK) and c-Jun N-terminal kinase (JNK), is activated in response to oxidative stress and modulates the balance between pro-apoptotic proteins such as Bcl-2-associated X protein (BAX) and anti-apoptotic proteins such as B-cell lymphoma 2 (Bcl-2), ultimately influencing mitochondrial integrity and caspase-3 activation [12,13,14]. Moreover, oxidative stress can suppress cAMP response element-binding protein (CREB), a key factor in neuronal survival and plasticity [15].

Given the critical role of oxidative stress and apoptosis in neurodegenerative diseases, natural compounds with antioxidant and anti-apoptotic properties have received considerable attention as potential neuroprotective agents [16,17]. Anthocyanins, a subclass of flavonoids found in various plant species, exhibit potent antioxidant, anti-inflammatory, and neuroprotective effects [18,19]. *Morus alba* L., commonly known as mulberry fruit, is particularly rich in anthocyanins and has demonstrated promising pharmacological activities, including free radical scavenging and protection against neuronal injury [11,20]. Vitamin C (ascorbic acid) was selected as a complementary compound because it functions as a water-soluble antioxidant capable of neutralizing ROS and supporting enzymatic antioxidant defenses [21,22]. In addition, it stabilizes anthocyanins against degradation and oxidation during first-pass metabolism, thereby enhancing their bioavailability. Moreover, physiological transport mechanisms enable vitamin C to reach the central nervous system, underscoring its relevance not only for neuroprotection in vitro and in vivo but also for potential clinical applications and practical use in functional foods and nutraceuticals [23]. Despite these complementary properties, the synergistic potential of anthocyanins and vitamin C in mitigating oxidative stress-induced neuronal injury remains largely unexplored, providing a clear rationale for the present study.

Therefore, the present study aimed to investigate the combined neuroprotective effects of anthocyanin-enriched *Morus alba* L. extract and vitamin C in SH-SY5Y human neuroblastoma cells exposed to H_2_O_2_-induced oxidative stress. We specifically examined their effects on oxidative biomarkers, endogenous antioxidant enzyme activities, and key apoptotic signaling pathways, including p-ERK, ERK, CREB, Bcl-2, BAX, and cleaved caspase-3. Our findings highlight the synergistic action of these natural food-derived compounds in protecting neuronal cells and support their potential application as functional food ingredients for the dietary management or prevention of oxidative stress-related neurodegenerative disorders.

## 2. Materials and Methods

### 2.1. Reagents and Chemicals

*Morus alba* L. (mulberry fruits var. Chiangmai) were obtained from the Queen Sirikit Department of Sericulture Center, located at 296 Moo 13, Ban That Subdistrict, Phen District, Udon Thani Province, Thailand (17.02411° N, 102.99020° E). A voucher specimen was deposited under the reference number 61,001 for authentication and future reference. To minimize variability of the plant matrix, all samples were collected from the same geographical location and harvested at a similar stage of ripeness and season. Vitamin C (ascorbic acid; C_6_H_8_O_6_; MW: 176.12; CAS No. 50-81-7), MTT reagent, dimethyl sulfoxide (DMSO), and the Immobilon^®^ Forte Western HRP substrate were purchased from Merck KGaA (Darmstadt, Germany).

All cell culture reagents, including Dulbecco’s Modified Eagle Medium (DMEM), fetal bovine serum (FBS), penicillin-streptomycin, non-essential amino acids (NEAA), CM-H_2_DCFDA (5-(and-6)-carboxy-2′,7′-dichlorofluorescein diacetate), and N-PER™ neuronal protein extraction buffer, were obtained from Thermo Fisher Scientific (Waltham, MA, USA).

For Western blotting, SDS-polyacrylamide gels, nitrocellulose membranes, and 0.1% Tween-20 were supplied by Bio-Rad (Hercules, CA, USA). All primary and secondary antibodies, including those specific to p-ERK, ERK, CREB, Bcl-2, BAX, cleaved caspase-3, and β-actin, were obtained from Cell Signaling Technology (Danvers, MA, USA).

### 2.2. Extraction of Anthocyanin-Enriched Morus alba L. Extract

The dried fruits of *Morus alba* L. were thoroughly cleaned and oven-dried at 60 °C for 48 h (Memmert GmbH, Schwabach, Germany) to prevent microbial growth and preserve heat-sensitive phytochemicals. The dried samples were then ground into a fine powder using a laboratory grinder. To obtain an anthocyanin-enriched extract, the powdered material was macerated in a 50% hydroalcoholic solution (ethanol/water, *v*/*v*) at room temperature for 72 h with occasional stirring. This solvent system was selected to enhance the extraction efficiency of polar phenolic compounds, particularly anthocyanins, while maintaining compound stability due to the mild temperature and absence of strong acids or bases.

After maceration, the mixture was centrifuged at 3000 rpm for 10 min to separate the solids, and the supernatant was filtered through Whatman No. 1 filter paper. The filtrate was concentrated using a rotary evaporator under reduced pressure at 40 °C (Heidolph Instruments GmbH, Schwabach, Germany) to remove ethanol. The remaining aqueous extract was then freeze-dried (lyophilized) to preserve bioactive compounds and obtain a stable powdered extract for further analysis. The extraction method was adapted from previously established protocols for anthocyanin isolation from pigmented plant sources [24].

The anthocyanin-enriched *Morus alba* L. extract used in this study was obtained from the same voucher specimen (No. 61001) as previously characterized in our earlier work [11,24]. Total anthocyanin content was determined using the pH differential spectrophotometric method, with absorbance measured at 520 and 700 nm in buffer systems at pH 1.0 and 4.5, and expressed as cyanidin-3-glucoside (C3G) equivalents. The extract contained a total anthocyanin content of 270.33 ± 4.19 µg C3G per mg of extract [11]. Consistent with previous chromatographic data, C3G was identified as the predominant anthocyanin, as confirmed by the HPLC profile obtained from the same botanical source at the Queen Sirikit Department of Sericulture Center, Udon Thani Province [11,24,25]. In addition, external studies have demonstrated that *Morus alba* L. typically exhibits high anthocyanin concentrations, with C3G consistently reported as the major component [26]. These findings support the consistency of our extract’s phytochemical characteristics with established anthocyanin profiles in *Morus alba* L. and reinforce its reliability for biological evaluation.

### 2.3. Determination of Optimal Ratio and Synergistic Effects of Morus alba L. Extract Combined with Vitamin C

Non-toxic concentrations of *Morus alba* L. extract and vitamin C were first determined using SH-SY5Y cell viability assays. *Morus alba* L. extract and vitamin C were tested at 0–1000 µg/mL, and concentrations just below their cytotoxic thresholds were selected for subsequent experiments: 31.25 µg/mL for *Morus alba* L. extract and 62.5 µg/mL for vitamin C, maintaining >80% cell viability.

To evaluate synergistic effects, the half-maximal effective concentrations (EC_50_) of *Morus alba* L. extract (21.30 µg/mL) and vitamin C (43.30 µg/mL) were determined and used as reference doses (1X). Combination treatments were prepared at 0.125X, 0.25X, 0.5X, 1X, and 2X relative to these EC_50_ values.

The interactions between *Morus alba* L. extract and vitamin C were analyzed using the SynergyFinder platform (version R-3.10.3.), based on the Loewe additivity model, suitable for compounds with overlapping mechanisms of action. The original dose–response data used as input for the analysis are provided in Appendix A. SynergyFinder generates a dose–response matrix and calculates quantitative synergy scores, which are visualized as 2D contour plots and 3D landscapes, allowing identification of concentrations that exhibit synergistic interactions.

To confirm the synergistic effect and select the optimal ratios, the Combination Index (CI) was calculated using the Chou–Talalay equation:CI = [D1/Dx1] + [D2/Dx2] where D1 and D2 are the doses of *Morus alba* L. extract and vitamin C in combination required to achieve a given effect, and Dx1 and Dx2 are the doses of the same agents alone required to achieve the same effect. CI values indicate the type of interaction:•CI < 1: synergism•CI = 1: additive effect•CI > 1: antagonism

Based on the synergy analysis and CI values, the 1X:1X and 1X:0.5X combinations (*Morus alba* L. extract: vitamin C) exhibited the strongest synergistic neuroprotective effects against H_2_O_2_-induced cytotoxicity. These combinations were designated as MAC1 (1X:1X) and MAC2 (1X:0.5X) for subsequent experiments.

### 2.4. Cell Culture and Treatment

Cell culture and treatment were performed as previously described in our established protocol. SH-SY5Y neuroblastoma cells (ATCC, CRL-2266) were maintained in DMEM supplemented with 10% fetal bovine serum, 1% penicillin-streptomycin, and 1% non-essential amino acids at 37 °C in a humidified incubator with 5% CO_2_. For experiments, cells were pretreated with varying concentrations of the combination for 24 h, followed by exposure to 200 µM H_2_O_2_ for 24 h with or without the test compounds. This concentration and exposure duration were selected based on preliminary dose–response experiments to induce moderate oxidative stress without excessive cell death, allowing evaluation of neuroprotective effects [11,27].

### 2.5. Cell Viability Assay

Cell viability was assessed using the MTT assay as previously reported [28]. Briefly, after treatment, MTT reagent was added to each well, and the plates were incubated to allow for formazan crystal formation. The crystals were then solubilized in DMSO, and absorbance was measured at 570 nm. Cell viability was expressed as a percentage relative to the control.

### 2.6. Protein Measurement

Total protein concentration in SH-SY5Y cell lysates was determined using the NanoDrop™ 2000/2000c spectrophotometer (Thermo Fisher Scientific). Briefly, cell lysates were appropriately diluted with buffer, and the absorbance at 280 nm was measured according to the manufacturer’s instructions. Each sample was analyzed in triplicate to ensure accuracy and reproducibility. Protein concentrations were calculated directly from the absorbance readings and used to normalize the results of lipid peroxidation, antioxidant enzyme activity assays, and Western blot analysis. NanoDrop was selected for its small sample requirement, rapid direct measurement, and compatibility with downstream assays.

### 2.7. Evaluation of Lipid Peroxidation

Lipid peroxidation was assessed by measuring MDA levels using a modified thiobarbituric acid reactive substances (TBARS) assay, as detailed in our previous study [29]. Briefly, cell lysates were processed, and the absorbance of the MDA–TBA complex was measured at 532 nm. Results were calculated based on a standard curve and expressed as nanograms of MDA per milligram of protein.

### 2.8. Evaluation of Intracellular Reactive Oxygen Species

Intracellular ROS levels were determined using the fluorescent probe CM-H_2_DCFDA, as described in our previous study [30]. Briefly, SH-SY5Y cells were treated with test compounds, incubated with CM-H_2_DCFDA, and then exposed to H_2_O_2_. Fluorescence intensity was measured at 488 nm excitation and 520 nm emission to assess ROS accumulation.

### 2.9. Assessment of Antioxidant Enzyme Activities

To evaluate cellular antioxidant defense mechanisms, the activities of CAT, SOD, and GSH-Px were measured in SH-SY5Y cell lysates, following methods previously described in our study [31].

CAT activity was assessed based on its ability to decompose H_2_O_2_, with the remaining H_2_O_2_ quantified via reaction with potassium permanganate and absorbance measured at 490 nm.

SOD activity was determined by its capacity to inhibit the reduction in cytochrome C in a xanthine–xanthine oxidase system, with absorbance monitored at 415 nm.

GSH-Px activity was measured through the rate of glutathione (GSH) oxidation in the presence of H_2_O_2_, followed by the reaction with DTNB, and absorbance was recorded at 412 nm.

Enzyme activities were calculated using standard curves prepared with purified enzymes and expressed as units per milligram of protein.

### 2.10. Western Blotting Analysis

Western blotting was performed to examine key proteins involved in apoptosis and survival pathways, including p-ERK, ERK, CREB, Bcl-2, BAX, and cleaved caspase-3, as described in detail in our previous study [32]. SH-SY5Y cells were lysed in protein extraction buffer, and total protein concentrations were determined. Equal amounts of protein were separated by SDS-PAGE, transferred to polyvinylidene fluoride (PVDF) membranes, and probed with specific primary antibodies. After incubation with HRP-conjugated secondary antibodies, protein bands were detected using chemiluminescence and quantified with Image Lab software. Protein expression levels were normalized to β-actin and expressed relative to the control group.

### 2.11. Statistical Analysis

Data were expressed as mean ± SEM. Group comparisons were analyzed using one-way ANOVA followed by Tukey’s post hoc test. Prior to ANOVA, the assumptions of normality and homogeneity of variance were evaluated using the Shapiro–Wilk test and Levene’s test, respectively. A *p*-value < 0.05 was considered statistically significant. All analyses were performed using SPSS Statistics v21.0 (IBM Corp., Armonk, NY, USA).

## 3. Results

### 3.1. Determination of Non-Toxic Doses of Morus alba L. Extract and Vitamin C

Figure 1 shows the effects of various concentrations of *Morus alba* L. extract and vitamin C (0–1000 µg/mL) on SH-SY5Y neuronal cell viability. Exposure to *Morus alba* L. extract significantly reduced cell viability at 62.5 µg/mL (*p* < 0.01 vs. control) and at higher concentrations. Similarly, vitamin C significantly decreased cell viability starting at 125 µg/mL (*p* < 0.001 vs. control) and at subsequent higher doses.

For subsequent experiments, concentrations just below the cytotoxic threshold were selected, specifically 31.25 µg/mL for *Morus alba* L. extract and 62.5 µg/mL for vitamin C. These doses maintained greater than 80% cell viability, indicating they are safe and suitable for treatment in SH-SY5Y cells.

### 3.2. Determination of EC_50_ Values for Morus alba L. Extract and Vitamin C

After identifying the optimal non-toxic concentrations of *Morus alba* L. extract (31.25 µg/mL) and vitamin C (62.5 µg/mL), we next determined the half-maximal effective concentration (EC_50_) of each compound in order to establish fixed ratios for combination treatment and evaluation of potential synergistic effects. Dose–response curves were generated using concentration ranges of 0–31.25 µg/mL for *Morus alba* L. extract and 0–62.5 µg/mL for vitamin C, with inhibitory activity assessed as percentage inhibition. As shown in Figure 2, the EC_50_ value of *Morus alba* L. extract was calculated to be 21.30 µg/mL (R^2^ = 0.994), while that of vitamin C was 43.30 µg/mL (R^2^ = 0.9948). These results provided the basis for selecting appropriate dose ratios to evaluate the synergistic neuroprotective effects of the combined treatment in subsequent experiments.

### 3.3. Evaluation of Synergistic Effects of Morus alba L. Extract and Vitamin C

To determine whether *Morus alba* L. extract and vitamin C act synergistically, combination treatments were analyzed using the SynergyFinder platform based on the Loewe additivity model, which is commonly applied to compounds with overlapping mechanisms of action. The EC_50_ values were 21.30 µg/mL for *Morus alba* L. extract and 43.30 µg/mL for vitamin C, which were used as the reference doses (1X). Additional concentrations were tested at 0.125X, 0.25X, 0.5X, 1X, and 2X relative to these EC_50_ values.

Figure 3A shows a two-dimensional synergy contour plot, while Figure 3B presents the corresponding three-dimensional landscape. The analysis yielded a Loewe synergy score of 22.488, indicating a strong synergistic interaction between *Morus alba* L. extract and vitamin C in protecting SH-SY5Y cells against H_2_O_2_-induced cytotoxicity. The highest synergy was observed around the concentrations corresponding to their respective EC_50_ values, demonstrating that the combined treatment provides greater protection than either compound alone.

### 3.4. Identification of Synergistic Dose Combinations of Morus alba L. Extract and Vitamin C

To identify the optimal synergistic doses of *Morus alba* L. extract and vitamin C, combination index (CI) values were calculated (Table 1), and the dose–response matrix was analyzed (Figure 4). The results showed that the 0.5X:0.5X, 1X:0.5X, and 1X:1X combinations exhibited synergistic interactions. Among these, the 1X:0.5X and 1X:1X combinations provided the highest protective effects, with values of 90.15 and 91.34, respectively.

Based on these findings, the 1X:0.5X and 1X:1X ratios were selected for further investigation of the combined neuroprotective effects of anthocyanin-enriched *Morus alba* L. extract and vitamin C (MAC) against H_2_O_2_-induced oxidative stress in SH-SY5Y cells. These combinations were designated as MAC1 (1X:1X) and MAC2 (1X:0.5X), as both CI values and the dose–response matrix consistently confirmed their strong synergistic efficacy.

### 3.5. Determination of Neuroprotective Effects of MAC

Following the identification of the two optimal ratios of anthocyanin-enriched *Morus alba* L. extract and vitamin C (MAC), their neuroprotective effects were evaluated. Figure 5A shows the morphological changes in SH-SY5Y neuronal cells under different treatment conditions. Cells exposed to H_2_O_2_ with vehicle exhibited a marked reduction in density, along with evident morphological damage, including cell shrinkage, shortened neurites, nuclear condensation, and the formation of apoptotic bodies. These alterations were accompanied by a significant decrease in cell viability (*p* < 0.001 vs. control group).

In contrast, cells treated with H_2_O_2_ in combination with either MAC1 or MAC2 displayed markedly preserved morphology and significantly higher viability compared with the H_2_O_2_ + vehicle group (all *p* < 0.001). These findings indicate that both MAC1 and MAC2 effectively attenuate H_2_O_2_-induced cytotoxicity in SH-SY5Y neuronal cells.

### 3.6. Effects of MAC on ERK/CREB Signaling Pathway

To explore the possible underlying mechanism of the neuroprotective effects of MAC, we focused on the ERK/CREB signaling pathway, which plays a critical role in neuronal survival and plasticity. The results presented in Figure 6 demonstrate that exposure to H_2_O_2_ with vehicle significantly reduced ERK expression (Figure 6B) and CREB expression (Figure 6C) compared with the control group (*p* < 0.01 and *p* < 0.001, respectively). In contrast, treatment with MAC1 and MAC2 markedly increased ERK levels (all *p* < 0.01 vs. H_2_O_2_ + vehicle) and CREB levels (all *p* < 0.001 vs. H_2_O_2_ + vehicle).

These findings suggest that MAC protects SH-SY5Y neuronal cells, at least in part, through activation of the ERK/CREB signaling pathway.

### 3.7. Effect of MAC on Bcl-2 and BAX Expression

Bcl-2 and BAX are key regulators of apoptosis, where Bcl-2 acts as an anti-apoptotic protein and BAX functions as a pro-apoptotic counterpart. The ERK/CREB signaling pathway has been reported to modulate the expression of these apoptotic regulators, thereby influencing neuronal survival. As shown in Figure 7, exposure to H_2_O_2_ with vehicle significantly reduced Bcl-2 expression (Figure 7B) while markedly increasing BAX expression (Figure 7C) compared with the control group (all *p* < 0.001). In contrast, treatment with MAC1 markedly upregulated Bcl-2 and downregulated BAX levels (all *p* < 0.001 vs. H_2_O_2_ + vehicle). Similarly, MAC2 also elevated Bcl-2 expression (*p* < 0.05 vs. H_2_O_2_ + vehicle) and suppressed BAX expression (*p* < 0.001 vs. H_2_O_2_ + vehicle), suggesting that MAC attenuates apoptosis through the regulation of Bcl-2 and BAX, potentially mediated by ERK/CREB signaling.

In addition to individual expression changes, the Bcl-2/BAX ratio was evaluated as an indicator of cellular susceptibility to apoptosis. As shown in Figure 8, the control group exhibited a ratio of 1.61 ± 0.05, consistent with a pro-survival profile. Exposure to H_2_O_2_ with vehicle treatment significantly reduced this ratio to 0.65 ± 0.03 (*p* < 0.001 vs. control), indicating a strong shift toward pro-apoptotic signaling. Treatment with MAC1 restored the ratio to 1.23 ± 0.07 (*p* < 0.001 vs. H_2_O_2_ + vehicle), while MAC2 increased it to 1.04 ± 0.05 (*p* < 0.05 vs. H_2_O_2_ + vehicle). Although both treatments improved the ratio compared with H_2_O_2_ alone, MAC1 exerted a more pronounced effect, suggesting greater efficacy in re-establishing the balance between pro- and anti-apoptotic proteins.

### 3.8. Effect of MAC on Cleaved Caspase-3 Expression

Cleaved caspase-3 is a critical executioner in the apoptotic pathway, activated downstream of mitochondrial regulators such as Bcl-2 and BAX. As illustrated in Figure 9, exposure to H_2_O_2_ with vehicle markedly increased cleaved caspase-3 expression compared with the control group (*p* < 0.001). In contrast, treatment with both MAC1 and MAC2 significantly reduced cleaved caspase-3 levels (all *p* < 0.05 vs. H_2_O_2_ + vehicle). These results suggest that MAC attenuates H_2_O_2_-induced apoptotic cell death by suppressing caspase-3 activation, further reinforcing its anti-apoptotic effects through modulation of the ERK/CREB signaling pathway and regulation of the Bcl-2/BAX balance.

### 3.9. Effect of MAC on Antioxidant Enzyme Activities

Oxidative stress plays a central role in triggering neuronal apoptosis, in part through activation of signaling pathways such as ERK/CREB and modulation of mitochondrial proteins including Bcl-2, BAX, and cleaved caspase-3. To further clarify the antioxidant contribution of MAC, the activities of CAT, SOD, and GSH-Px were evaluated (Figure 10). Exposure to H_2_O_2_ with vehicle significantly decreased the activities of CAT, SOD, and GSH-Px compared with the control group (*p* < 0.001, *p* < 0.001, and *p* < 0.01, respectively). In contrast, treatment with MAC1 markedly restored CAT, SOD, and GSH-Px activities (*p* < 0.01, *p* < 0.001, and *p* < 0.05 vs. H_2_O_2_ + vehicle, respectively). Similarly, MAC2 also enhanced CAT, SOD, and GSH-Px activities (*p* < 0.01, *p* < 0.001, and *p* < 0.05 vs. H_2_O_2_ + vehicle, respectively).

These findings indicate that MAC protects neuronal cells not only by inhibiting apoptosis but also by reinforcing the endogenous antioxidant defense system, thereby reducing oxidative stress and preserving neuronal integrity under H_2_O_2_-induced injury.

### 3.10. Effect of MAC on Intracellular ROS Production

Since excessive accumulation of ROS is a major driver of oxidative stress and downstream apoptotic signaling, we next assessed intracellular ROS levels in SH-SY5Y neuronal cells (Figure 11). Cells exposed to H_2_O_2_ with vehicle showed a marked increase in ROS generation compared with the control group (*p* < 0.001). Treatment with both MAC1 and MAC2 significantly attenuated ROS accumulation (all *p* < 0.001 vs. H_2_O_2_ + vehicle).

These results are consistent with the enhanced activities of antioxidant enzymes observed earlier, supporting the notion that MAC alleviates oxidative damage by reducing ROS overproduction, thereby contributing to its neuroprotective effects against H_2_O_2_-induced cytotoxicity.

### 3.11. Effect of MAC on Lipid Peroxidation (MDA Levels)

To further evaluate oxidative damage, we measured MDA, a marker of lipid peroxidation (Figure 12). Exposure to H_2_O_2_ with vehicle resulted in a significant elevation of MDA levels compared with the control group (*p* < 0.001). In contrast, treatment with either MAC1 or MAC2 markedly reduced MDA accumulation (all *p* < 0.001 vs. H_2_O_2_ + vehicle).

These findings, together with the observed reductions in ROS and enhancements in antioxidant enzyme activities, indicate that MAC effectively mitigates oxidative damage by limiting lipid peroxidation and restoring redox homeostasis in SH-SY5Y neuronal cells.

## 4. Discussion

Oxidative stress is one of the most important contributors to neurodegenerative diseases, including Alzheimer’s disease and Parkinson’s disease, where oxidative stress-induced mitochondrial dysfunction and dysregulated apoptotic signaling converge to drive progressive neuronal loss [33,34]. Our model using H_2_O_2_-induced oxidative stress is a classical paradigm that reproduces these mechanisms. In SH-SY5Y cells, H_2_O_2_ exposure led to excessive ROS accumulation, elevated lipid peroxidation, and suppression of endogenous antioxidant defenses, accompanied by activation of apoptotic pathways through Bcl-2/BAX imbalance, cytochrome c release, and cleaved caspase-3, together with dysregulation of the ERK/CREB signaling pathway. These results are in line with previous studies demonstrating that oxidative stress simultaneously promotes mitochondrial apoptosis and inhibits survival signaling, thereby accelerating neuronal injury.

H_2_O_2_-induced ROS accumulation caused mitochondrial dysfunction by disrupting the mitochondrial membrane potential and altering the balance of pro- and anti-apoptotic proteins [35]. Under physiological conditions, a high Bcl-2/BAX ratio stabilizes the mitochondrial outer membrane, preventing permeabilization. Conversely, when BAX exceeds Bcl-2, the membrane becomes destabilized, leading to the release of cytochrome c into the cytosol. Once released, cytochrome c binds to apoptotic protease-activating factor-1 (Apaf-1), promoting apoptosome formation. This multiprotein complex recruits and activates caspase-9, which in turn cleaves and activates executioner caspase-3. Activated caspase-3 initiates proteolytic degradation of structural and regulatory proteins, culminating in the execution phase of apoptosis [36,37]. Our results demonstrated that MAC treatment suppressed this cascade by restoring the Bcl-2/BAX ratio, preventing cytochrome c release, and reducing caspase-3 activation. These findings are consistent with established models of mitochondrial apoptosis in neurodegeneration.

In addition to apoptosis, ROS accumulation disrupted the ERK/CREB signaling pathway, which plays a pivotal role in synaptic plasticity and neuronal resilience [38,39,40]. Normally, ERK activation phosphorylates CREB, enabling transcription of anti-apoptotic and survival-promoting genes, including Bcl-2 [41]. In our model, H_2_O_2_ exposure suppressed ERK/CREB activity, leading to reduced Bcl-2 expression and increased susceptibility to apoptosis. The downregulation of CREB impairs transcription of anti-apoptotic genes, tipping the balance toward pro-apoptotic signaling. In this context, the decrease in Bcl-2 further destabilizes mitochondrial integrity, amplifying the apoptotic cascade [41,42]. Our results showed that MAC treatment reactivated ERK/CREB signaling, enhanced Bcl-2 expression, and thereby interrupted the feed-forward cycle of ROS-induced apoptosis. These observations support the idea that modulation of ERK/CREB is a critical mechanism by which MAC confers neuroprotection.

H_2_O_2_-induced ROS also promoted lipid peroxidation, a critical process in neurodegeneration [43,44]. Peroxidized lipids not only compromise membrane integrity but also generate toxic aldehydes such as MDA, which propagate oxidative damage by amplifying protein misfolding, DNA damage, and mitochondrial injury [45,46]. These effects further exacerbate mitochondrial apoptosis, as described above. Moreover, oxidative stress impaired enzymatic antioxidant defenses, as indicated by decreased activities of CAT, SOD, and GSH-Px, which accelerated lipid peroxidation and redox imbalance [43,47,48]. Our results revealed that MAC treatment significantly enhanced these endogenous antioxidant enzymes, reduced lipid peroxidation, and restored redox homeostasis. By reinforcing antioxidant defenses, MAC indirectly protected mitochondrial membranes, thereby limiting apoptotic activation and supporting neuronal survival.

Interestingly, the results showed that anthocyanin-enriched *Morus alba* L. extract and vitamin C exhibit strong synergism, as confirmed by both the combination index and the Loewe additivity model, which quantitatively demonstrated that the combined effect exceeded the predicted additive effect of each agent alone. Our results further indicate that the equimolar combination MAC1 (1X:1X) provides stronger neuroprotective effects than the sub-equimolar MAC2 (1X:0.5X), reflecting key mechanistic differences between the two ratios [11,27,49,50]. MAC1 maintains an optimal balance between anthocyanins and vitamin C, allowing effective regeneration of oxidized anthocyanins by vitamin C and stabilization of vitamin C by anthocyanins, thereby sustaining the redox cycle [51]. In contrast, the lower vitamin C concentration in MAC2 may be insufficient to fully regenerate anthocyanins, reducing ROS scavenging, antioxidant enzyme activation, and apoptosis suppression. Furthermore, anthocyanins from *Morus alba* L. are structurally diverse molecules capable of scavenging multiple ROS, chelating transition metals, and interacting with membranes, but their maximal efficacy depends on adequate vitamin C levels. Under physiological conditions, the stability and bioavailability of both anthocyanins and vitamin C are limited by factors such as pH, metabolism, and enzymatic degradation, which further supports the use of equimolar combinations in future in vivo or clinical studies [52,53,54,55].

In a translational context, our findings indicate that anthocyanin-enriched *Morus alba* L. extract combined with vitamin C has strong synergistic potential, supporting its application in functional food formulations or dietary supplements aimed at protecting neurons from oxidative stress-related neurodegeneration and age-associated damage. Future studies should validate these effects in primary neuronal cultures and in vivo models, clarify pharmacokinetic properties and bioavailability, and evaluate interactions with other redox-sensitive pathways. Collectively, this work provides a foundation for developing next-generation functional foods that target oxidative stress, apoptosis, and mitochondrial dysfunction to preserve neuronal function and support overall brain health.

## 5. Conclusions

In conclusion, the combination of anthocyanin-enriched *Morus alba* L. extract and vitamin C (MAC) exerts potent synergistic neuroprotective effects against H_2_O_2_-induced oxidative stress in SH-SY5Y neuronal cells. MAC effectively reduces ROS accumulation, limits lipid peroxidation, enhances endogenous antioxidant enzyme activities, and suppresses apoptosis through modulation of the ERK/CREB signaling pathway and Bcl-2/BAX-caspase-3 axis, as shown in Figure 13. The superior efficacy of the MAC1 (1X:1X) ratio highlights the importance of optimal stoichiometric combinations to maximize complementary mechanisms of action. These findings support the concept that multi-targeted interventions combining dietary polyphenols and classical antioxidants can provide enhanced neuronal protection compared with single-agent approaches. Overall, this study identifies MAC as a promising nutraceutical candidate, paving the way for the development of functional foods aimed at preventing or mitigating oxidative stress-related neurodegenerative disorders.

## Figures and Tables

**Figure 1 foods-14-03630-f001:**
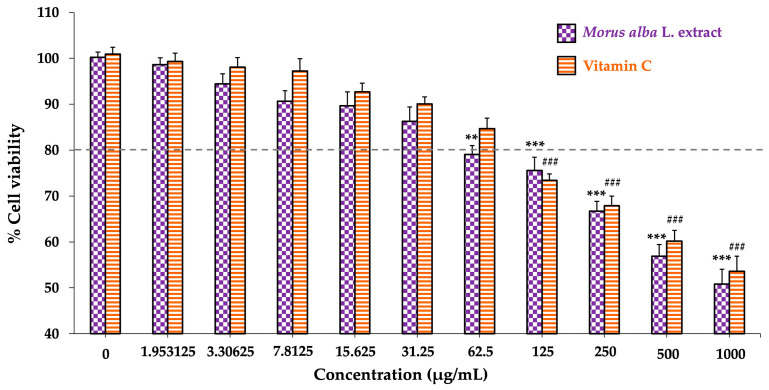
Effect of *Morus alba* L. extract and vitamin C on SH-SY5Y cell viability. Cells were treated with 0–1000 µg/mL of each compound for 24 h. Data are presented as mean ± SEM (*n* = 3). For *Morus alba* L. extract, ** *p* < 0.01 and *** *p* < 0.001 vs. control (0 µg/mL); for vitamin C, ^###^ *p* < 0.001 vs. control (0 µg/mL).

**Figure 2 foods-14-03630-f002:**
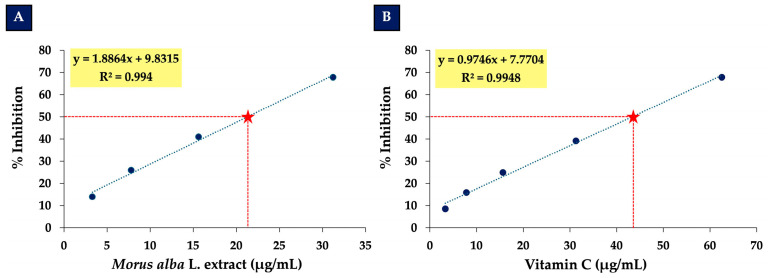
EC_50_ values for *Morus alba* L. extract and vitamin C. (**A**) Dose–response curve generated using 0–31.25 µg/mL for *Morus alba* L. extract. (**B**) Dose–response curve generated using 0–62.5 µg/mL for vitamin C.

**Figure 3 foods-14-03630-f003:**
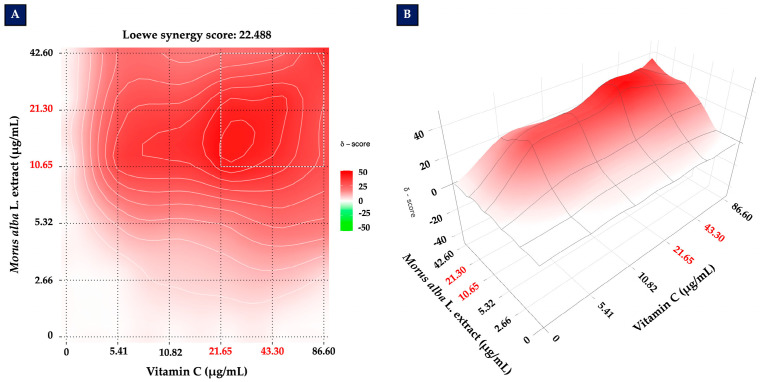
Synergistic effects of *Morus alba* L. extract and vitamin C. (**A**) Two-dimensional synergy contour plot and (**B**) three-dimensional synergy landscape generated using the SynergyFinder platform based on the Loewe additivity model. Concentrations were tested at 0.125X, 0.25X, 0.5X, 1X, and 2X relative to the EC_50_ values (21.30 µg/mL for *Morus alba* L. extract and 43.30 µg/mL for vitamin C).

**Figure 4 foods-14-03630-f004:**
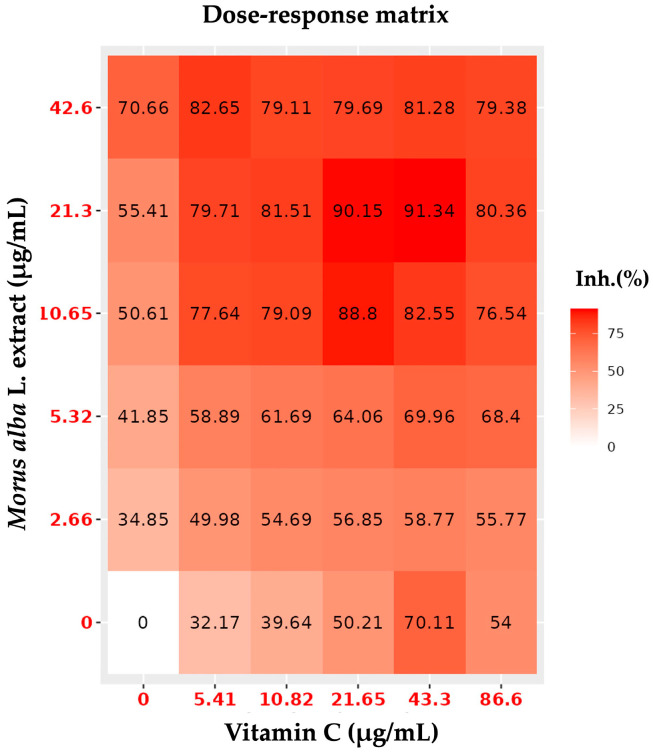
Dose–response matrix showing the synergistic interactions of *Morus alba* L. extract and vitamin C.

**Figure 5 foods-14-03630-f005:**
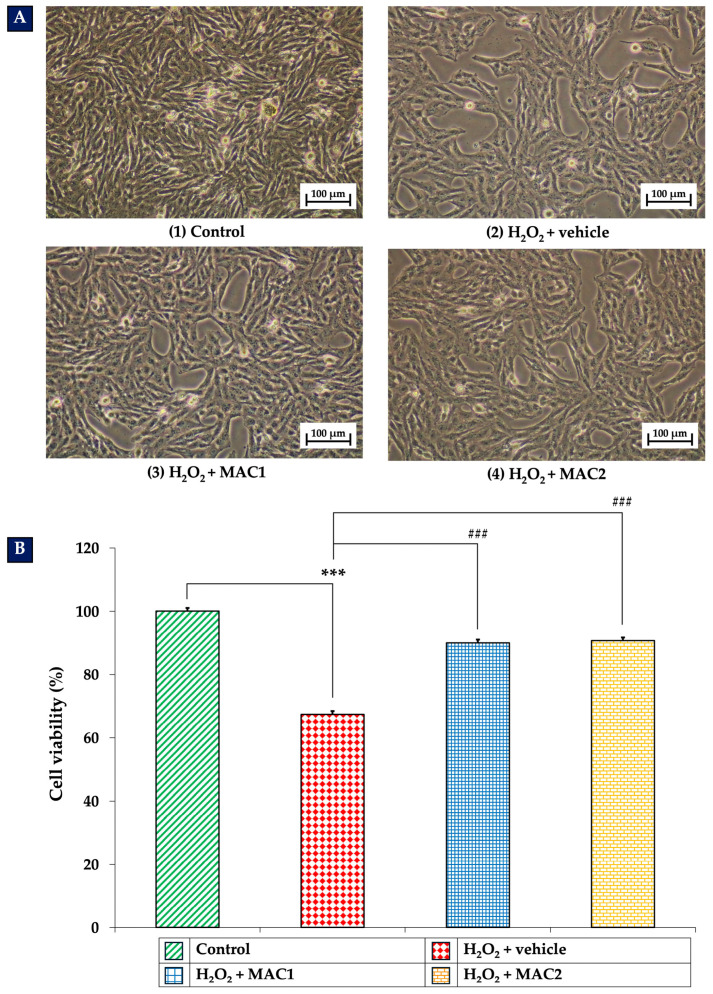
Neuroprotective effects of MAC on SH-SY5Y cells exposed to H_2_O_2_. (**A**) Micrographs illustrating cell morphology at 10× magnification. (**B**) Cell viability measurements expressed as mean ± SEM (*n* = 4). *** *p* < 0.001 vs. control; ^###^ *p* < 0.001 vs. H_2_O_2_ + vehicle. H_2_O_2_: hydrogen peroxide; MAC1: anthocyanin-enriched *Morus alba* L. extract combined with vitamin C at 1X:1X; MAC2: anthocyanin-enriched *Morus alba* L. extract combined with vitamin C at 1X:0.5X.

**Figure 6 foods-14-03630-f006:**
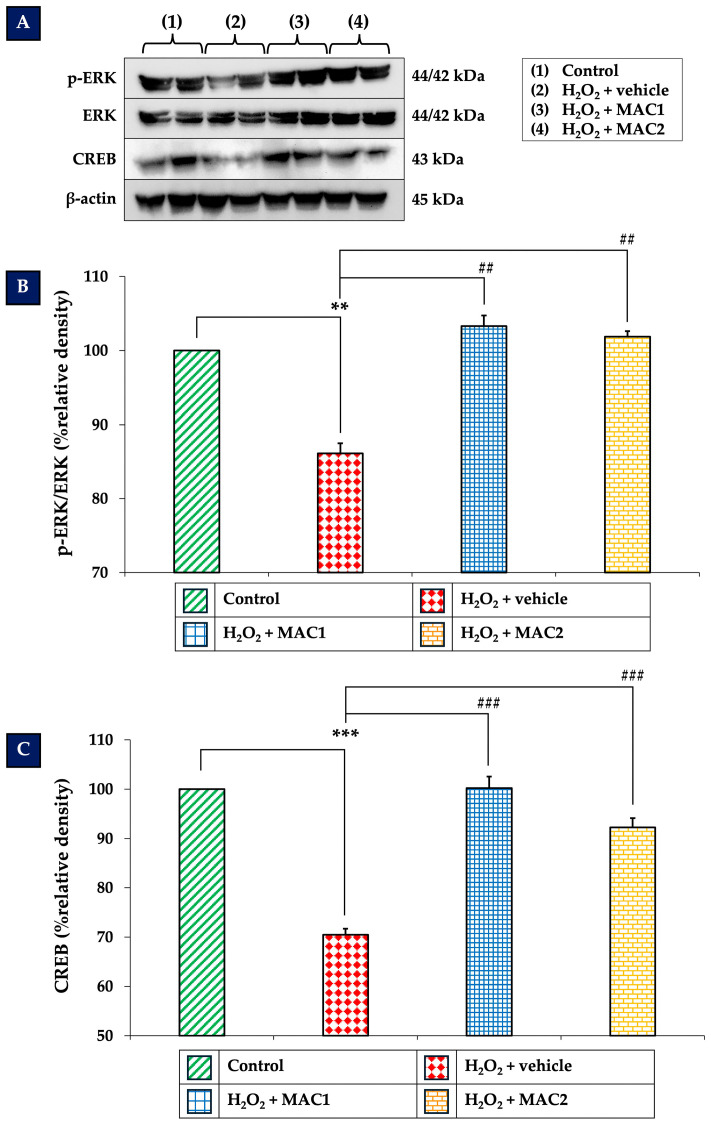
Effects of MAC on the ERK/CREB signaling pathway in SH-SY5Y cells exposed to H_2_O_2_. (**A**) Representative Western blots of ERK and CREB. (**B**,**C**) Quantification of ERK and CREB expression, respectively. Data are means ± SEM (*n* = 4). ** *p* < 0.01 and *** *p* < 0.001 vs. control; ^##^ *p* < 0.01 and ^###^*p* < 0.001 vs. H_2_O_2_ + vehicle. H_2_O_2_: hydrogen peroxide; MAC1: anthocyanin-enriched *Morus alba* L. extract combined with vitamin C at 1X:1X; MAC2: anthocyanin-enriched *Morus alba* L. extract combined with vitamin C at 1X:0.5X.

**Figure 7 foods-14-03630-f007:**
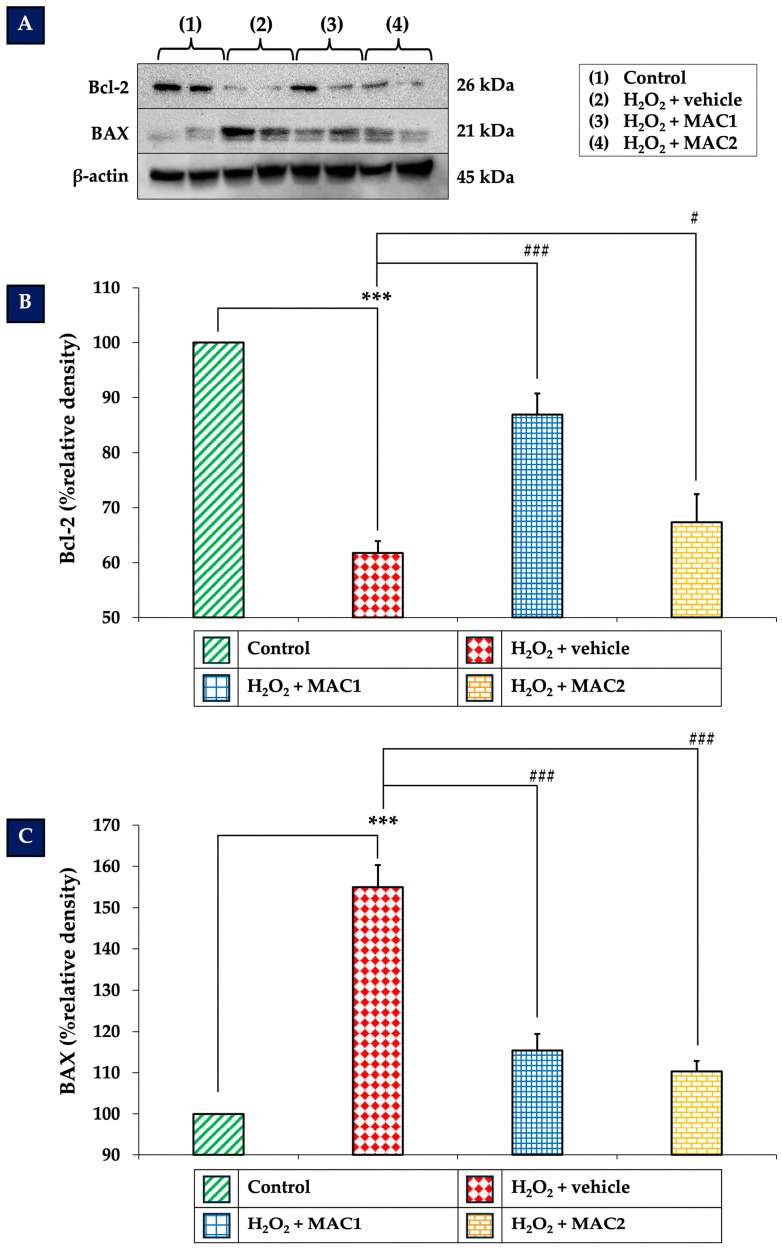
Effects of MAC on Bcl-2 and BAX expression in SH-SY5Y cells exposed to H_2_O_2_. (**A**) Representative Western blots of Bcl-2 and BAX. (**B**,**C**) Quantification of Bcl-2 and BAX expression, respectively. Data are means ± SEM (*n* = 4). *** *p* < 0.001 vs. control; ^#^ *p* < 0.05 and ^###^ *p* < 0.001 vs. H_2_O_2_ + vehicle. H_2_O_2_: hydrogen peroxide; MAC1: anthocyanin-enriched *Morus alba* L. extract combined with vitamin C at 1X:1X; MAC2: anthocyanin-enriched *Morus alba* L. extract combined with vitamin C at 1X:0.5X.

**Figure 8 foods-14-03630-f008:**
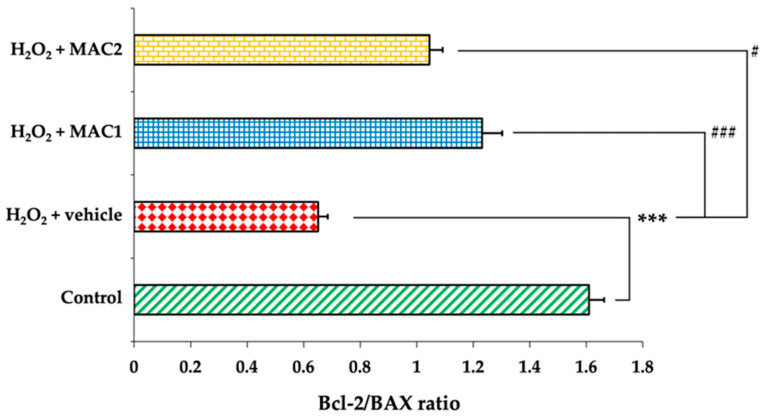
Effects of MAC on the Bcl-2/BAX ratio in SH-SY5Y cells exposed to H_2_O_2_. Data are means ± SEM (*n* = 4). *** *p* < 0.001 vs. control; ^#^ *p* < 0.05 and ^###^ *p* < 0.001 vs. H_2_O_2_ + vehicle. H_2_O_2_: hydrogen peroxide; MAC1: anthocyanin-enriched *Morus alba* L. extract combined with vitamin C at 1X:1X; MAC2: anthocyanin-enriched *Morus alba* L. extract combined with vitamin C at 1X:0.5X.

**Figure 9 foods-14-03630-f009:**
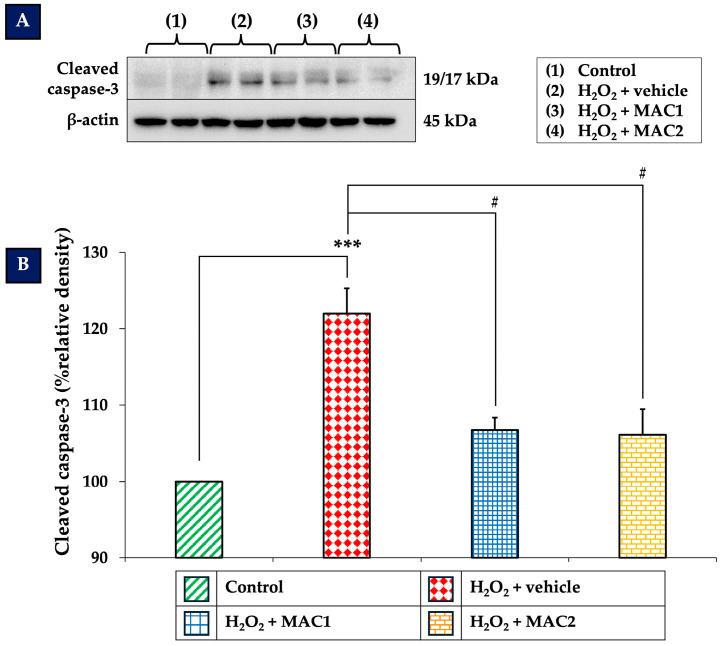
Effects of MAC on cleaved caspase-3 expression in SH-SY5Y cells exposed to H_2_O_2_. (**A**) Representative Western blots of cleaved caspase-3. (**B**) Quantification of cleaved caspase-3 expression. Data are means ± SEM (*n* = 4). *** *p* < 0.001 vs. control; ^#^ *p* < 0.05 vs. H_2_O_2_ + vehicle. H_2_O_2_: hydrogen peroxide; MAC1: anthocyanin-enriched *Morus alba* L. extract combined with vitamin C at 1X:1X; MAC2: anthocyanin-enriched *Morus alba* L. extract combined with vitamin C at 1X:0.5X.

**Figure 10 foods-14-03630-f010:**
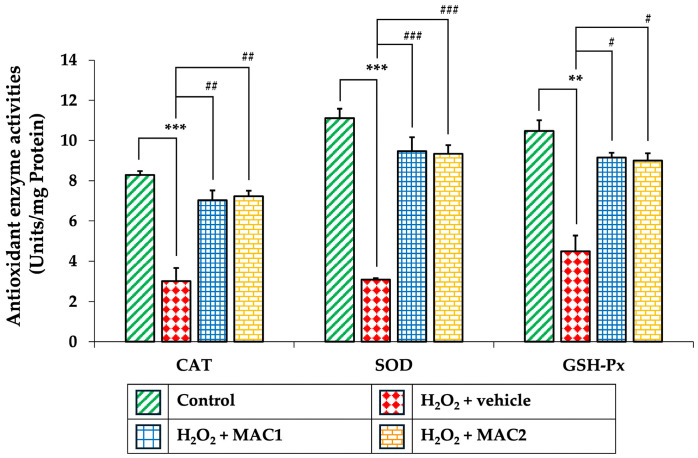
Effects of MAC on antioxidant enzyme activities in SH-SY5Y cells exposed to H_2_O_2_. Data are means ± SEM (*n* = 4). ** *p* < 0.01 and *** *p* < 0.001 vs. control; ^#^ *p* < 0.05, ^##^ *p* < 0.01 and ^###^ *p* < 0.001 vs. H_2_O_2_ + vehicle. H_2_O_2_: hydrogen peroxide; MAC1: anthocyanin-enriched *Morus alba* L. extract combined with vitamin C at 1X:1X; MAC2: anthocyanin-enriched *Morus alba* L. extract combined with vitamin C at 1X:0.5X; CAT: catalase; SOD: superoxide dismutase; GSH-Px: glutathione peroxidase.

**Figure 11 foods-14-03630-f011:**
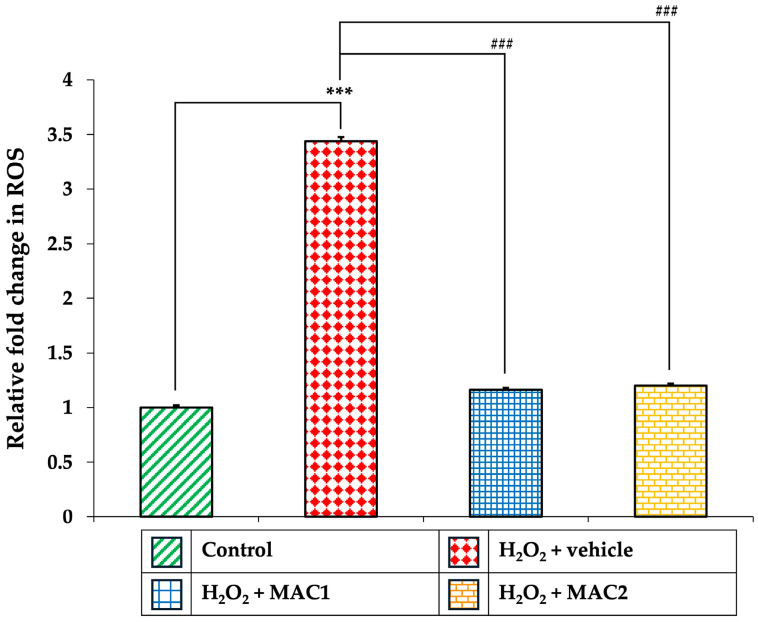
Effects of MAC on intracellular ROS production in SH-SY5Y cells exposed to H_2_O_2_. Data are means ± SEM (*n* = 4). *** *p* < 0.001 vs. control; ^###^
*p* < 0.001 vs. H_2_O_2_ + vehicle. H_2_O_2_: hydrogen peroxide; MAC1: anthocyanin-enriched *Morus alba* L. extract combined with vitamin C at 1X:1X; MAC2: anthocyanin-enriched *Morus alba* L. extract combined with vitamin C at 1X:0.5X; ROS: reactive oxygen species.

**Figure 12 foods-14-03630-f012:**
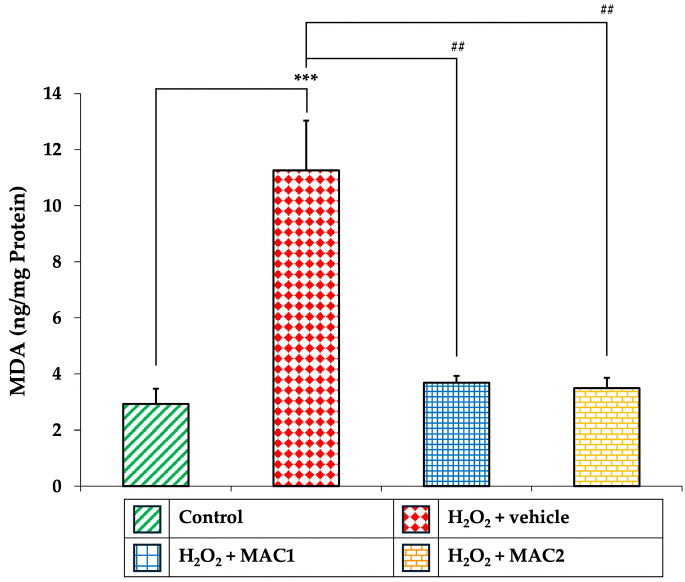
Effects of MAC on MDA levels in SH-SY5Y cells exposed to H_2_O_2_. Data are means ± SEM (*n* = 4). *** *p* < 0.001 vs. control; ^##^ *p* < 0.01 vs. H_2_O_2_ + vehicle. H_2_O_2_: hydrogen peroxide; MAC1: anthocyanin-enriched *Morus alba* L. extract combined with vitamin C at 1X:1X; MAC2: anthocyanin-enriched *Morus alba* L. extract combined with vitamin C at 1X:0.5X; MDA: malondialdehyde.

**Figure 13 foods-14-03630-f013:**
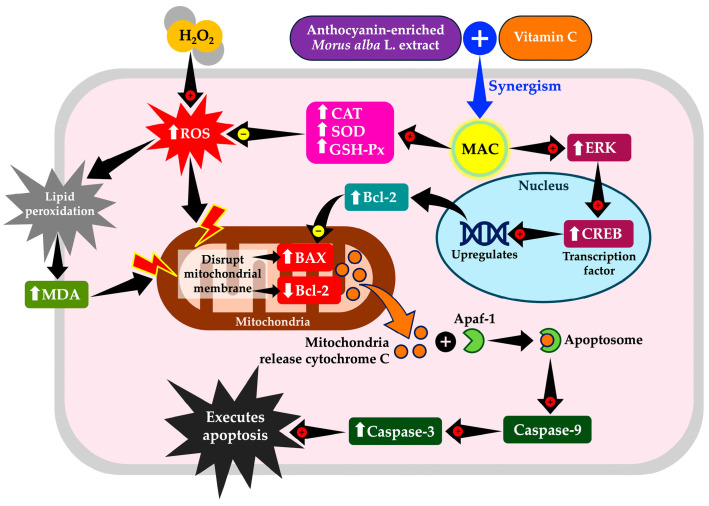
Schematic diagram summarizing the synergistic neuroprotective effects of anthocyanin-enriched *Morus alba* L. extract combined with vitamin C against H_2_O_2_-induced oxidative stress in SH-SY5Y neuronal cells. H_2_O_2_: hydrogen peroxide; ROS: reactive oxygen species; MDA: malondialdehyde; CAT: catalase; SOD: superoxide dismutase; GSH-Px: glutathione peroxidase; ERK: extracellular signal-regulated kinase; CREB: cAMP response element-binding protein; Bcl-2: B-cell lymphoma 2; BAX: Bcl-2-associated X; Apaf-1: apoptotic protease activating factor 1; MAC: anthocyanin-enriched *Morus alba* L. extract combined with vitamin C.

**Table 1 foods-14-03630-t001:** Combination index values of *Morus alba* L. extract and vitamin C.

Ratios (Dose Level)		% Inhibition	Combination Index Value	Interaction
*Morus alba* L. Extract	Vitamin C			
0.125X	0.125X	50.03 ± 0.61	1.58 ± 0.05	Antagonism
0.25X	0.125X	41.11 ± 1.79	1.43 ± 0.06	Moderate antagonism
0.5X	0.125X	22.36 ± 0.71	0.90 ± 0.03	Nearly additive effect
1X	0.125X	20.29 ± 0.69	1.16 ± 0.04	Slight antagonism
2X	0.125X	17.35 ± 0.82	1.01 ± 0.05	Nearly additive effect
0.125X	0.25X	45.31 ± 0.62	1.41 ± 0.03	Moderate antagonism
0.25X	0.25X	38.31 ± 1.08	1.36 ± 0.06	Moderate antagonism
0.5X	0.25X	20.91 ± 1.18	0.89 ± 0.06	Slight synergism
1X	0.25X	18.49 ± 1.07	1.04 ± 0.05	Nearly additive effect
2X	0.25X	20.89 ± 1.06	1.23 ± 0.07	Moderate antagonism
0.125X	0.5X	43.15 ± 0.47	1.34 ± 0.02	Moderate antagonism
0.25X	0.5X	35.94 ± 0.50	1.25 ± 0.02	Moderate antagonism
0.5X	0.5X	11.20 ± 0.58	0.46 ± 0.02	Synergism
1X	0.5X	9.85 ± 1.12	0.61 ± 0.08	Synergism
2X	0.5X	20.31 ± 1.48	1.23 ± 0.11	Moderate antagonism
0.125X	1X	41.23 ± 1.32	1.28 ± 0.04	Moderate antagonism
0.25X	1X	30.04 ± 0.57	1.05 ± 0.02	Nearly additive effect
0.5X	1X	17.45 ± 0.87	0.74 ± 0.04	Moderate synergism
1X	1X	8.66 ± 0.98	0.52 ± 0.06	Synergism
2X	1X	18.73 ± 0.73	1.09 ± 0.05	Nearly additive effect
0.125X	2X	44.23 ± 0.82	1.37 ± 0.03	Moderate antagonism
0.25X	2X	31.60 ± 2.02	1.11 ± 0.07	Slight antagonism
0.5X	2X	23.46 ± 1.91	1.02 ± 0.09	Nearly additive effect
1X	2X	19.64 ± 1.37	1.12 ± 0.08	Slight antagonism
2X	2X	20.63 ± 1.29	1.17 ± 0.07	Slight antagonism

Data are presented as means ± SEM (*n* = 3).

## Data Availability

All data generated or analyzed in this study are included in the article and its Appendix A. Further information can be obtained from the corresponding authors upon request.

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
