# Peer review of "Synergistic Effects of Anthocyanin-Enriched *Morus alba* L. Extract and Vitamin C: Promising Nutraceutical Ingredients in Functional Food Development for Neuroprotection"

_foods, 2025, doi:10.3390/foods14213630_

Round 1

Reviewer 1 Report

Comments and Suggestions for Authors

The authors present a very interesting study on the interaction between anthocyanins and vitamin C and propose a mechanism of action based on a cellular model. The work is well organized and structured; however, the following aspects should be considered: 
Although the extraction methods and the composition of the anthocyanins present have been previously reported, an analysis of the total anthocyanin content and/or a chromatographic analysis could also be included. could provide insight into the variability of the plant matrix.
Regarding the EC₅₀ information and the potential synergistic or antagonistic actions, as well as the dose–response curves, the authors should present the main results in the main manuscript and place some of them in the supplementary material, in order to give more space to the results related to the mechanisms of action.
Improve the equation format, including number

Author Response

Response to reviewer and editor suggestion

We sincerely thank you for your letter and the reviewers’ insightful comments on our manuscript, Synergistic Effects of Anthocyanin-Enriched Morus alba L. Extract and Vitamin C: Promising Nutraceutical Ingredients in Functional Food Development for Neuroprotection (Manuscript ID: foods-3936406).

We greatly appreciate the opportunity to revise our manuscript and are grateful for the constructive feedback. We apologize for any errors in the initial submission and acknowledge the reviewers’ invaluable input, which has helped enhance the scientific rigor and clarity of our work.

We have carefully considered each comment and made revisions accordingly. Below, we provide a detailed account of the main corrections and our responses to the reviewers’ suggestions.

Response to reviewer 1

The authors present a very interesting study on the interaction between anthocyanins and vitamin C and propose a mechanism of action based on a cellular model. The work is well organized and structured; however, the following aspects should be considered:

Comments 1: Although the extraction methods and the composition of the anthocyanins present have been previously reported, an analysis of the total anthocyanin content and/or a chromatographic analysis could also be included

Response 1: We thank the reviewer for this valuable suggestion. In response, we have revised the Materials and Methods section (2.2) to include additional quantitative and compositional details of the Morus alba L. extract. The total anthocyanin contents were determined using the pH differential spectrophotometric method and expressed as cyanidin-3-glucoside (C3G) equivalents (Mairuae et al., 2023). Furthermore, consistent with previous chromatographic findings, C3G was identified as the predominant anthocyanin, as confirmed by the HPLC profile reported in a related study utilizing the same botanical source from the Queen Sirikit Department of Sericulture Center, Udon Thani Province (Wattanathorn et al., 2019 & Palachai et al., 2019).

In addition to our previously published data, we have incorporated external literature demonstrating that Morus alba L. typically exhibit high anthocyanin concentrations, with C3G frequently representing the major component. For example, a comparative analysis of 12 Morus cultivars revealed that C3G accounted for approximately 67% of total anthocyanins, reaching levels up to 19.51 mg/g dry weight, while another HPLC-based study identified C3G and cyanidin-3-rutinoside (C3R) as the principal anthocyanins (Chen et al., 2022; Chen et al., 2006; Qin et al., 2010). Together, these findings support the consistency of our extract’s composition with established anthocyanin profiles in Morus alba L. and reinforce the reliability of the bioactive composition used in this study.

References:

  • Mairuae, N.; Palachai, N.; Noisa, P. The Neuroprotective Effects of the Combined Extract of Mulberry Fruit and Mulberry Leaf against Hydrogen Peroxide-Induced Cytotoxicity in SH-SY5Y Cells. BMC Complement. Med. Ther. 2023, 23, 117. https://doi.org/10.1186/s12906-023-03930-z.
  • Palachai, N.; Wattanathorn, J.; Muchimapura, S.; Thukham-Mee, W. Antimetabolic Syndrome Effect of Phytosome Containing the Combined Extracts of Mulberry and Ginger in an Animal Model of Metabolic Syndrome. Oxid. Med. Cell Longev. 2019, 2019, 5972575. https://doi.org/10.1155/2019/5972575.
  • Wattanathorn, J.; Kawvised, S.; Thukham-Mee, W. Encapsulated Mulberry Fruit Extract Alleviates Changes in an Animal Model of Menopause with Metabolic Syndrome. Oxid. Med. Cell. Longev. 2019, 2019, 5360560. https://doi.org/10.1155/2019/5360560.
  • Chen, T.; Shuang, F.-F.; Fu, Q.-Y.; Ju, Y.-X.; Zong, C.-M.; Zhao, W.-G.; Zhang, D.-Y.; Yao, X.-H.; Cao, F.-L. Evaluation of the Chemical Composition and Antioxidant Activity of Mulberry (Morus alba L.) Fruits from Different Varieties in China. Molecules 202227, 2688. https://doi.org/10.3390/molecules27092688
  • Chen, P.-N.; Chu, S.-C.; Chiou, H.-L.; Kuo, W.-H.; Chiang, C.-L.; Hsieh, Y.-S. Mulberry Anthocyanins, Cyanidin 3-Rutinoside and Cyanidin 3-Glucoside, Exhibited an Inhibitory Effect on the Migration and Invasion of a Human Lung Cancer Cell Line. Cancer Lett. 2006, 235, 248–259. https://doi.org/10.1016/j.canlet.2005.04.033
  • Qin, C.; Li, Y.; Niu, W.; Ding, Y.; Zhang, R.; Shang, X. Analysis and Characterisation of Anthocyanins in Mulberry Fruit. Czech J. Food Sci. 2010, 28, 117–126. https://doi.org/10.17221/228/2008-CJFS

Comments 2: Could provide insight into the variability of the plant matrix.

Response 2: We sincerely thank the reviewer for this valuable comment. To address this, we have revised the Materials and Methodssection to provide precise information on the source and collection of the Morus alba L. The fruits were obtained from the Queen Sirikit Department of Sericulture Center, located at 296 Moo 13, Ban That Subdistrict, Phen District, Udon Thani Province, Thailand, along Mittraphap Road (Udon Thani – Nong Khai route) (17.02411° N, 102.99020° E), and a voucher specimen (No. 61001) was deposited for authentication.

To minimize variability of the plant matrix, all samples were collected from the same geographical location and harvested at a similar stage of ripeness and season. These clarifications ensure consistency of the plant material and provide readers with transparency regarding the reproducibility and reliability of the extract used in this study.

Comments 3: Regarding the EC₅₀ information and the potential synergistic or antagonistic actions, as well as the dose–response curves, the authors should present the main results in the main manuscript and place some of them in the supplementary material, in order to give more space to the results related to the mechanisms of action.

Response 3: We sincerely thank the reviewer for this constructive suggestion. In response, we have revised the manuscript to present the key findings of the synergistic effects in the main text, while providing detailed input data in the Supplementary Materials for transparency and reproducibility. Specifically:

  • In Section 3.3 (Evaluation of Synergistic Effects of Morus alba L. Extract and Vitamin C), we highlighted that the analysis yielded a Loewe synergy score of 22.488, indicating a strong synergistic interaction between the two compounds.
  • In Section 3.4 (Identification of Synergistic Dose Combinations of Morus alba L. Extract and Vitamin C), we reported that the 1X:0.5X and 1X:1X combinations provided the highest protective effects, with values of 15 and 91.34, respectively.

The original dose–response data used for SynergyFinder analysis are provided in Supplementary Table S1. The main manuscript summarizes the key synergy results, including the Loewe synergy score and the most effective dose combinations, while the supplementary data allow readers to examine the full input dataset and reproduce the analysis if desired.

Comments 4: Improve the equation format, including number.

Response 4: We thank the reviewer for this suggestion. We have reviewed the equations in the manuscript. As per the journal’s standard procedures, the production team will adjust the formatting and numbering of all equations upon acceptance to ensure consistency with the journal style.

Thank you once again for your valuable feedback. We appreciate the time and effort invested by the reviewers and editor in evaluating our manuscript. We have carefully addressed each point raised and made necessary revisions accordingly. We eagerly await further feedback and guidance from the editorial team.

Yours sincerely,

Nut Palachai

Reviewer 2 Report

Comments and Suggestions for Authors

The manuscript presented investigates the synergistic neuroprotective effects of an anthocyanin-enriched Morus alba L. extract, in combination with vitamin C, against H₂O₂-induced oxidative stress in SH-SY5Y neuroblastoma cells. The topic is of great current interest, given the growing focus on nutraceuticals for the prevention of neurodegenerative diseases. However, it also has some weaknesses that need to be addressed appropriately, which I list below.

1. Why use vitamin C as a potential molecule with a neuroprotective effect? According to its ADMET properties, it is unable to cross the blood-brain barrier and therefore, in a real situation, it could not reach the brain to exert its pharmacological effect. 
2. The introduction could benefit from a brief mention of why the combination of anthocyanins and vitamin C was chosen, beyond their individual antioxidant properties. It could help by answering the question: Is there a biochemical basis for expecting synergy?
3. The manuscript refers to an extract of Morus alba L. enriched with anthocyanins, but no phytochemical analysis of the extract is provided. It is essential to know the composition of the extract, especially the concentration of the main anthocyanins and other phenolic compounds. Without this characterization, the reproducibility of the study is questionable, and it is impossible to attribute the observed effects specifically to anthocyanins. I suggest supplementing the work with HPLC-MS or similar analysis.
4.    The methods section mentions treatment with H2O2, but the concentration and exposure time are not specified.
5. Why was NanoDrop used to quantify proteins and not the Bradford method, which is the standard in these cases?
6. It is mentioned that the ANOVA test was used, but it is not specified whether the statistical assumptions for applying this test were met. 
7. Why does the control group in Figures 6B and 6C have no dispersion? Was only one test performed? The same applies to Figure 7.
8. There are approximately 7 self-citations, which seems excessive for a total of 52 citations. 

Author Response

Response to reviewer and editor suggestion

We sincerely thank you for your letter and the reviewers’ insightful comments on our manuscript, Synergistic Effects of Anthocyanin-Enriched Morus alba L. Extract and Vitamin C: Promising Nutraceutical Ingredients in Functional Food Development for Neuroprotection (Manuscript ID: foods-3936406).

We greatly appreciate the opportunity to revise our manuscript and are grateful for the constructive feedback. We apologize for any errors in the initial submission and acknowledge the reviewers’ invaluable input, which has helped enhance the scientific rigor and clarity of our work.

We have carefully considered each comment and made revisions accordingly. Below, we provide a detailed account of the main corrections and our responses to the reviewers’ suggestions.

Response to reviewer 2

The manuscript presented investigates the synergistic neuroprotective effects of an anthocyanin-enriched Morus alba L. extract, in combination with vitamin C, against H₂O₂-induced oxidative stress in SH-SY5Y neuroblastoma cells. The topic is of great current interest, given the growing focus on nutraceuticals for the prevention of neurodegenerative diseases. However, it also has some weaknesses that need to be addressed appropriately, which I list below.

Comments 1 Why use vitamin C as a potential molecule with a neuroprotective effect? According to its ADMET properties, it is unable to cross the blood-brain barrier and therefore, in a real situation, it could not reach the brain to exert its pharmacological effect.

Response 1: We thank the reviewer for this insightful comment. Vitamin C was selected as a candidate for combination with anthocyanin-enriched Morus alba L. extract for several reasons:

  1. Central nervous system and clinical relevance: While the conventional view suggests limited passive diffusion of vitamin C across the blood-brain barrier (BBB), physiological transport systems facilitate its uptake into the CNS. The reduced form, ascorbate, is actively transported into neurons and glial cells via SVCT2 transporters, and the oxidized form, dehydroascorbic acid (DHA), crosses the BBB through GLUT1 transporters and is subsequently reduced back to ascorbate in the brain. These mechanisms ensure that vitamin C can reach the CNS and exert neuroprotective effects, highlighting its potential relevance in clinical settings and practical applications.
  2. In vitro mechanistic studies: In SH-SY5Y neuronal cells, vitamin C can directly scavenge reactive oxygen species and modulate apoptosis, allowing us to evaluate its synergistic interaction with anthocyanins under controlled conditions.
  3. Functional food rationale: For potential oral intake via functional foods or nutraceuticals, anthocyanins are prone to degradation and oxidation in the gastrointestinal tract. Vitamin C can stabilize anthocyanins and enhance their antioxidant effects, thereby improving overall neuroprotective potential and practical utility in food formulations.

We have added a brief explanation in the Introduction to clarify the rationale for using vitamin C, emphasizing its mechanistic role in vitro, its physiological and clinical relevance in the CNS, and its practical utility in functional food and nutraceutical development.

References:

  • Agus, D. B., Gambhir, S. S., Pardridge, W. M., Spielholz, C., Baselga, J., Vera, J. C., & Golde, D. W. (1997). Vitamin C crosses the blood-brain barrier in the oxidized form through the glucose transporters. The Journal of clinical investigation100(11), 2842–2848. https://doi.org/10.1172/JCI119832
  • Harrison, F. E., & May, J. M. (2009). Vitamin C function in the brain: vital role of the ascorbate transporter SVCT2. Free radical biology & medicine46(6), 719–730. https://doi.org/10.1016/j.freeradbiomed.2008.12.018
  • Huang, J., Agus, D. B., Winfree, C. J., Kiss, S., Mack, W. J., McTaggart, R. A., Choudhri, T. F., Kim, L. J., Mocco, J., Pinsky, D. J., Fox, W. D., Israel, R. J., Boyd, T. A., Golde, D. W., & Connolly, E. S., Jr (2001). Dehydroascorbic acid, a blood-brain barrier transportable form of vitamin C, mediates potent cerebroprotection in experimental stroke. Proceedings of the National Academy of Sciences of the United States of America98(20), 11720–11724. https://doi.org/10.1073/pnas.171325998
  • Gess, B., Sevimli, S., Strecker, J. K., Young, P., & Schäbitz, W. R. (2011). Sodium-dependent vitamin C transporter 2 (SVCT2) expression and activity in brain capillary endothelial cells after transient ischemia in mice. PloS one6(2), e17139. https://doi.org/10.1371/journal.pone.0017139
  • Kook, S. Y., Lee, K. M., Kim, Y., Cha, M. Y., Kang, S., Baik, S. H., Lee, H., Park, R., & Mook-Jung, I. (2014). High-dose of vitamin C supplementation reduces amyloid plaque burden and ameliorates pathological changes in the brain of 5XFAD mice. Cell death & disease5(2), e1083. https://doi.org/10.1038/cddis.2014.26

Comments 2 The introduction could benefit from a brief mention of why the combination of anthocyanins and vitamin C was chosen, beyond their individual antioxidant properties. It could help by answering the question: Is there a biochemical basis for expecting synergy?

Response 2: We thank the reviewer for this constructive comment. We have revised the Introduction to provide a clearer rationale for combining anthocyanins and vitamin C, beyond their individual antioxidant effects. The combination was chosen because vitamin C can stabilize anthocyanins against degradation and oxidation during first-pass metabolism, thereby enhancing their bioavailability and preserving their antioxidant capacity. Additionally, at the cellular level, anthocyanins and vitamin C act through complementary mechanisms to reduce reactive oxygen species, modulate apoptotic signaling, and support endogenous antioxidant enzyme systems. These overlapping but distinct biochemical pathways provide a mechanistic basis for expecting synergistic neuroprotective effects, which we have now highlighted in the revised Introduction.

Comments 3 The manuscript refers to an extract of Morus alba L. enriched with anthocyanins, but no phytochemical analysis of the extract is provided. It is essential to know the composition of the extract, especially the concentration of the main anthocyanins and other phenolic compounds. Without this characterization, the reproducibility of the study is questionable, and it is impossible to attribute the observed effects specifically to anthocyanins. I suggest supplementing the work with HPLC-MS or similar analysis.

Response 3: We thank the reviewer for this insightful comment. In response, we have updated the Materials and Methods section (2.2) to include detailed phytochemical characterization of the anthocyanin-enriched Morus alba L. extract used in this study. The extract was obtained from the same voucher specimen (No. 61001) as previously characterized in our earlier work (Mairuae, Palachai, & Noisa, 2023). The total anthocyanin content was determined using standard spectrophotometric methods, yielding values of 270.33 ± 4.19 µg cyanidin-3-glucoside (C3G) equivalents/mg extract, respectively.

Furthermore, consistent with previous chromatographic analyses, cyanidin-3-O-glucoside (C3G) was confirmed as the predominant anthocyanin, as demonstrated by the HPLC profile reported in a related study using the same botanical source from the Queen Sirikit Department of Sericulture Center, Udon Thani Province (Palachai et al., 2019; Wattanathorn et al., 2019). In addition, external studies have shown that Morus alba L. generally exhibit high anthocyanin content, with C3G frequently dominant among cultivars (Chen et al., 2022; Chen et al., 2006; Qin et al., 2010). These data collectively enhance transparency and reproducibility, supporting that the observed neuroprotective effects can be attributed, at least in part, to the anthocyanins present in the extract.

References:

  • Mairuae, N.; Palachai, N.; Noisa, P. The Neuroprotective Effects of the Combined Extract of Mulberry Fruit and Mulberry Leaf against Hydrogen Peroxide-Induced Cytotoxicity in SH-SY5Y Cells. BMC Complement. Med. Ther. 2023, 23, 117. https://doi.org/10.1186/s12906-023-03930-z.
  • Palachai, N.; Wattanathorn, J.; Muchimapura, S.; Thukham-Mee, W. Antimetabolic Syndrome Effect of Phytosome Containing the Combined Extracts of Mulberry and Ginger in an Animal Model of Metabolic Syndrome. Oxid. Med. Cell Longev. 2019, 2019, 5972575. https://doi.org/10.1155/2019/5972575.
  • Wattanathorn, J.; Kawvised, S.; Thukham-Mee, W. Encapsulated Mulberry Fruit Extract Alleviates Changes in an Animal Model of Menopause with Metabolic Syndrome. Oxid. Med. Cell. Longev. 2019, 2019, 5360560. https://doi.org/10.1155/2019/5360560.
  • Chen, T.; Shuang, F.-F.; Fu, Q.-Y.; Ju, Y.-X.; Zong, C.-M.; Zhao, W.-G.; Zhang, D.-Y.; Yao, X.-H.; Cao, F.-L. Evaluation of the Chemical Composition and Antioxidant Activity of Mulberry (Morus alba L.) Fruits from Different Varieties in China. Molecules 202227, 2688. https://doi.org/10.3390/molecules27092688
  • Chen, P.-N.; Chu, S.-C.; Chiou, H.-L.; Kuo, W.-H.; Chiang, C.-L.; Hsieh, Y.-S. Mulberry Anthocyanins, Cyanidin 3-Rutinoside and Cyanidin 3-Glucoside, Exhibited an Inhibitory Effect on the Migration and Invasion of a Human Lung Cancer Cell Line. Cancer Lett. 2006, 235, 248–259. https://doi.org/10.1016/j.canlet.2005.04.033
  • Qin, C.; Li, Y.; Niu, W.; Ding, Y.; Zhang, R.; Shang, X. Analysis and Characterisation of Anthocyanins in Mulberry Fruit. Czech J. Food Sci. 2010, 28, 117–126. https://doi.org/10.17221/228/2008-CJFS

Comments 4 The methods section mentions treatment with H2O2, but the concentration and exposure time are not specified.

Response 4: We thank the reviewer for pointing this out. We have revised the Methods section to specify that SH-SY5Y cells were exposed to 200 µM H₂O₂ for 24 hours following pretreatment with the test compounds. This concentration and exposure time were chosen based on preliminary experiments to induce sufficient oxidative stress while maintaining viable cells, enabling reliable assessment of the neuroprotective effects of anthocyanin-enriched Morus alba L. extract and vitamin C.

Comments 5 Why was NanoDrop used to quantify proteins and not the Bradford method, which is the standard in these cases?

Response 5: We thank the reviewer for this comment. We used the NanoDrop™ 2000/2000c spectrophotometer to quantify total protein because it provides several advantages in our experimental context compared with the Bradford assay:

  1. Minimal sample requirement: NanoDrop requires only 1–2 µL of sample, preserving valuable cell lysates for downstream assays such as lipid peroxidation, antioxidant enzyme activity measurements, and western blot analysis.
  2. Rapid and direct measurement: Protein concentration can be determined directly from absorbance at 280 nm without additional reagents or incubation steps, allowing high-throughput and consistent measurements.
  3. Reproducibility and accuracy for clean lysates: For relatively pure cell lysates without high concentrations of interfering substances (e.g., detergents, strong reducing agents), absorbance at 280 nm provides accurate quantification comparable to colorimetric assays like Bradford.
  4. Compatibility with downstream assays: Using NanoDrop avoids introducing dyes or chemicals that might interfere with enzymatic activity assays or western blot normalization.

We have added a brief justification in the Methods section to clarify this choice:

2.6. Protein Measurement

Total protein concentration in SH-SY5Y cell lysates was determined using the NanoDrop™ 2000/2000c spectrophotometer (Thermo Fisher Scientific). Briefly, cell lysates were appropriately diluted with buffer, and the absorbance at 280 nm was measured according to the manufacturer’s instructions. Each sample was analyzed in triplicate to ensure accuracy and reproducibility. Protein concentrations were calculated directly from the absorbance readings and used to normalize the results of lipid peroxidation, antioxidant enzyme activity assays, and western blot analysis. NanoDrop was selected for its small sample requirement, rapid direct measurement, and compatibility with downstream assays”

References:

  • Desjardins, P., & Conklin, D. (2010). NanoDrop microvolume quantitation of nucleic acids. Journal of visualized experiments : JoVE, (45), 2565. https://doi.org/10.3791/2565
  • Versmessen, N., Van Simaey, L., Negash, A. A., Vandekerckhove, M., Hulpiau, P., Vaneechoutte, M., & Cools, P. (2024). Comparison of DeNovix, NanoDrop and Qubit for DNA quantification and impurity detection of bacterial DNA extracts. PloS one19(6), e0305650. https://doi.org/10.1371/journal.pone.0305650

Comments 6 It is mentioned that the ANOVA test was used, but it is not specified whether the statistical assumptions for applying this test were met.

Response 6: We thank the reviewer for this important comment. We have clarified in the Methods section that the assumptions for ANOVA, including normality and homogeneity of variances, were evaluated prior to statistical analysis to ensure the validity and reliability of the results.

Specifically, the Shapiro–Wilk test was employed to assess the normality of data distribution within each group. This test compares the observed distribution of the data to a perfectly normal distribution and provides a W statistic and p-value; a p-value greater than 0.05 indicates that the data do not significantly deviate from normality. The Shapiro–Wilk test was chosen because it is one of the most powerful and widely accepted methods for detecting departures from normality, particularly suitable for small to moderate sample sizes commonly used in in vitro studies.

Levene’s test was applied to evaluate homogeneity of variances across groups, which is another key assumption for one-way ANOVA. This test assesses whether group variances are statistically equal by analyzing the absolute deviations of each observation from its group mean. A non-significant result (p > 0.05) indicates that the assumption of equal variances is satisfied.

By confirming that both normality and homogeneity assumptions were met, we ensured that the application of one-way ANOVA was appropriate and that subsequent multiple comparison tests (Tukey’s post hoc) provided valid and interpretable results.

Comments 7 Why does the control group in Figures 6B and 6C have no dispersion? Was only one test performed? The same applies to Figure 7.

Response 7: We thank the reviewer for this comment. Figures 6, 7, and 9 present relative protein expression levels, which are calculated based on the ratio of the band intensity of each sample to that of the control group. In this type of analysis, the control group is assigned a value of 100% as a reference point, and all treatment groups are expressed relative to this baseline. Because the control is used as the denominator for normalization, its relative value is fixed at 100%, resulting in no visible dispersion. This approach is a standard method in western blot quantification, as it accounts for inter-gel variability and allows direct comparison of fold changes in protein expression across experimental conditions.

Comments 8 There are approximately 7 self-citations, which seems excessive for a total of 52 citations.

Response 8 We thank the reviewer for this observation. After revision, the manuscript now contains 55 references, of which seven are self-citations, representing approximately 12.7% of the total. This proportion remains within the acceptable range for scientific publications. These citations were included because they provide essential background information and describe experimental methodologies directly relevant to the present work. Moreover, these methods are standardized and widely adopted in this research area, making their inclusion necessary to ensure clarity, reproducibility, and transparency. Excluding them could disrupt methodological continuity, limit interpretability of the findings, and potentially raise concerns regarding self-plagiarism, as these references document our previously published protocols and foundational work.

Thank you once again for your valuable feedback. We appreciate the time and effort invested by the reviewers and editor in evaluating our manuscript. We have carefully addressed each point raised and made necessary revisions accordingly. We eagerly await further feedback and guidance from the editorial team.

Yours sincerely,

Nut Palachai

Round 2

Reviewer 2 Report

Comments and Suggestions for Authors

I agree with the changes made to the manuscript and therefore recommend its acceptance.